# Rintatolimod Induces Antiviral Activities in Human Pancreatic Cancer Cells: Opening for an Anti-COVID-19 Opportunity in Cancer Patients?

**DOI:** 10.3390/cancers13122896

**Published:** 2021-06-09

**Authors:** Dana A. M. Mustafa, Lawlaw Saida, Diba Latifi, Leonoor V. Wismans, Willem de Koning, Lona Zeneyedpour, Theo M. Luider, Bernadette van den Hoogen, Casper H. J. van Eijck

**Affiliations:** 1Department of Pathology, The Tumor Immuno-Pathology (TIP) Laboratory, Erasmus University Medical Center, 3015 Rotterdam, The Netherlands; d.mustafa@erasmusmc.nl; 2Department of Surgery, The Tumor Immuno-Pathology (TIP) Laboratory, Erasmus University Medical Center, 3015 Rotterdam, The Netherlands; l.saida@erasmusmc.nl (L.S.); d.latifi@erasmusmc.nl (D.L.); l.wismans@erasmusmc.nl (L.V.W.); 3Clinical Bioinformatics Unit, Department of Pathology, The Tumor Immuno-Pathology (TIP) Laboratory, Erasmus University Medical Center, 3015 Rotterdam, The Netherlands; w.dekoning.1@erasmusmc.nl; 4Department of Neurology, Clinical and Cancer Proteomics, Erasmus University Medical Center, 3015 Rotterdam, The Netherlands; l.zeneyedpour@erasmusmc.nl (L.Z.); t.luider@erasmusmc.nl (T.M.L.); 5Department of Viroscience, Erasmus University Medical Center, 3015 Rotterdam, The Netherlands; b.vandenhoogen@erasmusmc.nl

**Keywords:** SARS-CoV-2, Rintatolimod, Toll-like receptor 3, epithelial cancerous cells, RNase L, interferon signaling, cytokines

## Abstract

**Simple Summary:**

Specific treatment for COVID-19 infections in cancer patients is lacking while the demand for treatment is increasing. Therefore, we explored the effect of Rintatolimod, a Toll-like receptor 3 (TLR3) agonist, on human epithelial cancerous cells. Our results demonstrated that Rintatolimod stimulated an anti-viral effect by producing RNase L that blocks virus replication. Moreover, Rintatolimod activated the innate and the adaptive immune systems by activating a cascade of actions in human cancerous cells. We believe that Rintatolimod should be considered in the treatment regimens of cancer patients who suffer from SARS-CoV-2 infection.

**Abstract:**

Severe acute respiratory virus-2 (SARS-CoV-2) has spread globally leading to a devastating loss of life. Large registry studies have begun to shed light on the epidemiological and clinical vulnerabilities of cancer patients who succumb to or endure poor outcomes of SARS-CoV-2. Specific treatment for COVID-19 infections in cancer patients is lacking while the demand for treatment is increasing. Therefore, we explored the effect of Rintatolimod (Ampligen^®^) (AIM ImmunoTech, Ocala, FL, USA), a Toll-like receptor 3 (TLR3) agonist, to treat uninfected human pancreatic cancer cells (HPACs). The direct effect of Rintatolimod was measured by targeted gene expression profiling and by proteomics measurements. Our results show that Rintatolimod induces an antiviral effect in HPACs by inducing RNase-L-dependent and independent pathways of the innate immune system. Treatment with Rintatolimod activated the interferon signaling pathway, leading to the overexpression of several cytokines and chemokines in epithelial cells. Furthermore, Rintatolimod treatment increased the expression of angiogenesis-related genes without promoting fibrosis, which is the main cause of death in patients with COVID-19. We conclude that Rintatolimod could be considered an early additional treatment option for cancer patients who are infected with SARS-CoV-2 to prevent the complicated severity of the disease.

## 1. Introduction

Since December 2019, severe acute respiratory virus-2 (SARS-CoV-2), the infection responsible for coronavirus disease-2019 (COVID-19), has spread globally leading to a devastating loss of life. On 11 March 2020, the World Health Organization (WHO) declared COVID-19 as a pandemic [1]. COVID-19 displays symptoms ranging from mild to severe (pneumonia) that can lead to death in some individuals [2,3,4]. SARS-CoV-2 is a typical single-stranded RNA virus of the family Coronaviridae, which also contains viruses such as SARS-CoV-1 and MERS-CoV. SARS-CoV-2, like SARS-CoV-1, enters epithelial, bronchial, and alveolar lung cells, and alveolar macrophages, by binding to the angiotensin-converting enzyme 2 (ACE2) receptor [5].

SARS-CoV-2 inhibits the anti-viral response by evading the innate immune system. When the virus moves towards the lower respiratory tract via airways, it triggers a strong immune response [6]. The host immune system recognizes the surface epitopes of the virus after it gains access inside the target cell. Macrophages, monocytes, and neutrophils increase the production of pro-inflammatory cytokines [7]. B cells recognize the N viral protein very quickly and produce antibodies against it, while the S viral protein requires 4–8 days to be recognized [8]. B cells produce IgA, IgG, and IgM antibodies at different time points in infected patients, which provide systemic immunity in different organs [9]. At the same time, infected cells recruit T cells that recognize the MHC class I that promotes cytotoxic T cells to kill the virus-infected cells [10]. The virus stimulates Th1/Th17 cells that lead to aggravated inflammatory responses and may result in “cytokine storms” that lead to immunopathology like pulmonary edema and pneumonia [6].

Patients who suffer from cancer and receive treatments may have an increased risk of developing an infectious disease [11]. Cancer patients often experience weight loss, or even cachexia in severe cases [12]. Cancer and its treatments can lead to poor nutrition in patients. Undernourishment weakens the quality of life and therapeutic response, and also affects the immune system [12]. Age, sex, and health condition, such as cardiovascular diseases and cancer, are linked to the severity of COVID-19 infections [13]. However, studies that took most high-risk factors into account found that having cancer in hospitalized patients was associated with an increased risk of dying compared to COVID-19 patients without cancer [14]. Various large registry studies have also provided inside clinical features associated with the risk of adverse COVID-19 outcomes in cancer patients, which show an increased vulnerability in cancer patients for SARS-CoV-2 with a subsequent poor outcome [15]. Evidence collected from national and international cancer registries shows that patients with cancer infected with SARS-CoV-2 have a higher probability of death compared with patients without cancer [15]. The key factors associated with the poor outcome are common factors like age, male sex, smoking history, number of comorbidities, and poor performance status. Viruses such as SARS-CoV-2 do not only infect normal epithelial cells but infect tumor cells as well due to the well-known impaired innate immunity and decreased anti-viral response of cancer cells [16].

Towards developing specific anti-SARS-CoV-2 treatment options, researchers have discovered various antiviral antibodies [17,18]. However, the usefulness of these treatments in infected cancer patients still under investigation. Other drugs like anti-malaria chloroquine showed a considerable anti-SARS-CoV-2 effect, despite the unpleasant side effects [19]. In addition, dexamethasone is used in patients hospitalized with COVID-19 receiving either invasive mechanical ventilation or oxygen alone. One of the most promising drugs to treat SARS-CoV-2 is remdesivir, which inhibits the viral RNA-dependent RNA polymerase [20]. Despite these advancements, an anti-SARS-CoV-2 drug that is used specifically to treat cancer patients who suffer from COVID-19 is still lacking.

One of the novel therapeutic options for cancer is immunotherapy, which has shown promising results. The immunotherapy strategy depends on boosting the patient’s immune system by reversing the tumor-antigen-specific T cell tolerance induced by the host tumor. Toll-like receptor (TLR) agonists, such as Rintatolimod, are currently under investigation as adjuvants in cancer immunotherapy clinical trials due to their ability to activate immune cells and selectively promote inflammation. In-depth analysis of the immune parameters in patients with pancreatic cancer reveals new insights into the interplay of these immune mechanisms and survival. It also provides the basic knowledge and the rationale for new (Immuno)therapeutic approaches and combination therapies [21].

Rintatolimod (Ampligen™) is an immunomodulatory mismatched double-stranded polymer of RNA (dsRNA) drug that is developed and used for the treatment of a variety of infections and even various types of cancer [22,23,24]. Rintatolimod has completed FDA-authorized Phase 1, 2, and 3 placebo-controlled clinical trials for chronic fatigue syndrome (CFS) [25]. In addition, Rintatolimod has been tested and showed promising results to treat severe acute respiratory syndrome (SARS) in a mouse model [26], prevent influenza virus infection [27], hepatitis C virus (HCV) [28], and treat human immunodeficiency virus (HIV) in clinical studies [29]. It is an optimized Toll-like receptor 3 (TLR3) agonist and endogenous interferon (IFN) inducer in late-stage clinical development, with the ability to augment both innate and acquired immunity including cellular responses, like T cell response, in humans with immunodeficiency (HIV disease) [30]. It has also induced natural killer (NK) and B cell responses in normal human volunteers [27,31].

Rintatolimod is approved for severe chronic fatigue syndrome in Argentina. Recently, the FDA and the EU granted orphan drug designations to Rintatolimod for the treatment of patients with pancreatic cancer [32]. The approval is based on the multiyear Early Access Program held at Erasmus University Medical Center, which indicated that Rintatolimod resulted in a statistically significant overall survival (OS) benefit in this patient population. The median OS with Rintatolimod was approximately 100% higher than the control group (In preparation). In addition, Rintatolimod was used to treat patients with metastasized colorectal cancer [33]. It was also used together with vaccine therapy to treat patients with stage II–IV human epidermal growth factor receptor 2 (HER2)-positive breast cancer [34].

Rintatolimod (Ampligen^®^) is a double-stranded RNA molecule (dsPoly (I: C12U)) that binds and activates TLR3, which is located intracellularly at the membrane of endosomes of various tissue types like placenta, pancreas, and airway epithelial cells [35]. TLR3 is also expressed on the surface of a broad range of innate immune cells and antigen-processing cells such as monocytes, dendritic cells (DCs), macrophages, NK cells, and T cells [23,24,36,37]. Furthermore, TLR3 is highly expressed in the human placenta and pancreas, and in epithelial cells of the oropharynx, esophagus, and lower airways [38,39]. Typically, TLR3 is activated by pathogens such as double-stranded (ds)RNA entering cells. TLR3 serves as a pathogen recognition receptor to stimulate the innate immune response against many respiratory pathogens [24,29,36]. This receptor is part of a family of pattern recognition receptors that detects pathogens before the activation of adaptive immunity. Therefore, TLR3 is critical in the first line of immunological defense against a broad spectrum of lethal viruses [24]. Taken together, we hypothesize that Rintatolimod could also be beneficial for cancer patients with early signs of COVID-19 infections.

The aim of this study was to investigate the effect of Rintatolimod in inducing Type I interferon in pancreatic cancer cells. Interferon induction is important in the anti-viral response, thereby inhibiting SARS-CoV-2 replication.

## 2. Materials and Methods

### 2.1. Cell Lines and Treatment

Two pancreatic cancer cell lines were used in the study: BxPC-3 and CFPAC-1. These cell lines were chosen because they are among the most studied human epithelial pancreatic cancer cell lines available. Cells were cultured using RPMI-1640 media supplemented with 25% FBS and 10% DMSO. Cells were treated overnight with three different concentrations of Rintatolimod: 0.5, 1.25, and 2.5 mg/mL, and with PBS as a control. These concentrations of Rintatolimod were chosen because they reflect the concentrations that were administered to pancreatic cancer patients (in preparation). Cell culturing and treatment were performed three times and were repeated independently for RNA and protein measurements. Cell pellets were collected, washed three times with PBS, and snapped frozen on dry ice and stored at −80 °C until the time of preparation.

### 2.2. RNA Extraction and Gene Expression Profile

An RNeasy FF Macro kit (Qiagen, Hilden, Germany) was used to extract RNA from the cell pellets following the manufacturer’s instructions. RNA quality and quantity were examined using the Agilent 2100 Bioanalyzer (Agilent Technologies, Santa Clara, CA, USA). RNA concentration was corrected to include the percentage of the sample that was >300 base pairs. Because interferon pathway-related genes are included in the PanCancer Immune profile panel of NanoString (Nanostring Technologies, Seattle, WA, USA) the panel was used to profile the samples. A total of 200 ng of RNA samples were hybridized with the panel probes at 65 °C for 17 h. RNA expression levels were measured using the nCounter^®^ FLEX Instrument (Nanostring Technologies). The counting of the genes was performed by scanning 490 fields of view (FOV).

### 2.3. Protein Extraction and Proteomic Profile

Cells were thawed and prepared to be measured by nano-liquid chromatography (nano-LC) as described previously [40]. In short, 0.1% RapiGest (Waters, Milford, MA, USA) in 50 mm NH_4_HCO_3_ pH 8.0 was used and the cells were disrupted by external sonication to ensure cell rupturing. For protein solubilization and denaturation, the samples were incubated at 37 and 100 °C for 5 and 15 min, respectively. Gold grade trypsin was used to fragment proteins into small peptides. After preparation, the protein extracted samples were diluted 1:10 and transferred to LC-vials and measured by nano-LC Orbitrap Fusion mass spectrometry (MS) (Thermo Fisher Scientific, San Jose, CA, USA). Raw mass spectrometer data files and tandem MS (MS/MS) spectra were extracted by Mascot Deamon (Matrix Science, London, UK). Peptide identifications were accepted if they exceeded a peptide probability of 95.0%. Protein identifications were accepted if protein probability exceeded 99.0% and at least two peptides were identified. Proteins that contained similar peptides and could not be differentiated based on MS/MS analysis alone were grouped. The mass spectrometry proteomics data have been deposited to the ProteomeXchange Consortium via the PRIDE [41] partner repository with the data set identifiers PXD026116 and 10.6019/PXD026116.

### 2.4. Data Analysis 

nSolver™ software (Nanostring Technologies) was used to analyze the expression profiles. The expression of 19 housekeeping genes was used to normalize the profiled data. A two-tailed Student’s t-test was used to determine the differentially expressed genes or proteins between cell lines that were not treated (controls, *n* = 2) and those that were treated with various concentrations of Rintatolimod (*n* = 6), with *p*-value ≤  0.01 being considered statistically significant. The heat maps were generated in R software (4.0.3).

## 3. Results

### 3.1. Rintatolimod Stimulated the Interferon Signaling Pathway in HPAC Cells 

Rintatolimod upregulated the expression of TLR3 in pancreatic cancer cells about 7 times more than the expression in untreated cells. Genes that are activated by TLR3 and involved in the innate immunity like Toll-like receptor adaptor molecule 1 (TICAM1) and TNF receptor associated factor 3 (TRIF3) were also found to be upregulated by Rintatolimod almost twice as much they were expressed in the untreated cells. In addition, the treated cells were found to overexpress STAT1, STAT2, and STAT3, 10, 2.7, and 1.9 times, respectively. Furthermore, treatment with Rintatolimod promoted the expression of interferon-induced protein with tetratricopeptide repeats 1 and 2 like IFIT1 and IFIT2, which can restrict virus replication (Figure 1).

### 3.2. Rintatolimod Produced Immunomodulatory Activity in HPAC Cells 

No elevated levels of IFN-α, IFN-β, or IFN-γ were observed upon treating pancreatic cancer cells with Rintatolimod. However, upregulation of various cytokines and chemokines was observed in the treated cells. CXCL10 was the highest overexpressed cytokine that showed an upregulation of almost 200 times upon treating the cancer cells with Rintatolimod. Other members of the C-X-C motif chemokine family were found to be upregulated in pancreatic cancer cells after Rintatolimod treatment, like CXCL 1, CXCL2, CXCL3, CXCL8 (IL-8), CXCL11, CXCL14, and CXCL16 (Figure 2). More than 33 cytokine-related genes were found to be overexpressed at least twice in pancreatic cancer cells treated with Rintatolimod as compared to untreated cells (Figure 2). Importantly, CXCL12 chemokine was not found to be upregulated by Rintatolimod.

### 3.3. Rintatolimod Induced the Expression of MHC Class I and II Histocompatibility in HPAC Cells

Antigen-presenting genes were overexpressed in cancerous cells that were treated with Rintatolimod. MHC class I, specifically HLA-B, HLA-C, and HLA-A showed overexpression of 50, 20, and 13 times, respectively, compared to untreated cells (Figure 3). In addition, antigen peptide transporting genes like TAP-1 and TAP-2 that mediate the unidirectional translocation of peptide antigens from the cytosol to the endoplasmic reticulum for loading onto MHC class I molecules were overexpressed after treating with Rintatolimod. Genes of MHC class II also showed an overexpression after treatment with Rintatolimod. HLA-DR, HLA-DP, HLA-DM, and HLA-DO showed overexpression between 13 and 7 times as compared to untreated cells (Figure 3).

### 3.4. Proteomics Measurements Validated the Antiviral Activities in HPAC Cells Treated with Rintatolimod

The functional effect of Rintatolimod was investigated on BxPC-3 pancreatic cancer cells. More than 200 proteins were found to be differentially expressed between treated and untreated BxPC-3 pancreatic cancer cells; 132 of them were overexpressed after treatment with Rintatolimod. The treatment induced the expression of 2′–5′-oligoadenylate synthase-like protein (OASL). Several OASL-related peptides were identified in cells that were treated with the highest concentration of Rintatolimod (Table 1). In addition, the treatment induced several ATP-related proteins like ATP-dependent RNA helicase DDX60 and DDX58. Both helicase proteins promote RIG-I-like receptor-mediated signaling that has an antiviral function. Moreover, proteomics data showed overexpression of interferon-induced proteins and HLA class I histocompatibility proteins, which validate our previous RNA findings.

## 4. Discussion

The aim of this study was to demonstrate the effect of Rintatolimod to create an antiviral effect in epithelial cancer cells. Our results highlighted that Rintatolimod modulates antiviral activities in epithelial cancer cells in various directions. Rintatolimod induced the expression of the 2′–5′-oligoadenylate synthase 3 (OAS3), which elicits antiviral activities through bioactivation of the ribonuclease L (RNase L) pathway. Activation of RNase L leads to degradation of cellular as well as viral RNA, resulting in the inhibition of protein synthesis and terminating the viral replication 28. In addition, Rintatolimod induced the expression of 2′–5′-oligoadenylate synthase-like protein (OASL) that activates an alternative antiviral pathway independent of RNase L.

In addition to degrading RNA, we showed that Rintatolimod induced the innate immune response in pancreatic cancer cells. Rintatolimod triggered TLR3, which activated TICAM1 and MyD88 and mediate NF-kB expression that subsequently activated transcription factors such as IRF1 and IRF7. Interferon Regulatory Factors activate STAT1, STAT2, and STAT3 that stimulate the innate and acquired immune responses, mediate cellular responses to interleukin, and regulate inflammatory responses to infection (Figure 4). Rintatolimod induced overexpression of STAT1 and STAT2 results in the expression of IFN-stimulated genes (ISG) that drive cells to an antiviral state, as reported before [30]. The detected overexpression of STAT3 is important, as STAT3 regulates inflammation by regulating the differentiation of naive CD4+ T cells into T helper cells or Regulatory T cells [31]. The STAT family genes regulate the expression of interferon-stimulated genes that drive cells to an antiviral state. Furthermore, treatment with Rintatolimod promoted the expression of interferon-induced protein with tetratricopeptide repeats 1 and 2 (IFIT1 and IFIT2), which can restrict virus replication through the alternation of protein synthesis and enhancing the antiviral proteins to bind to viral RNA directly [32]. We also detected the overexpression of various other genes with an antiviral effect that were enhanced after Rintatolimod treatment, like interferon gamma inducible protein 16 (IFI16), interferon alpha inducible protein (IFI27), and TICAM 1 (Figure 1). Treatment with Rintatolimod increased the expression of TLR3 significantly, especially in cells that express TLR3 at a low level like CFPAC1 cells (Figure 5). TLR3 sensing of Rintatolimod led to the overexpression of the TRIF3 gene that activates the TRIF signaling pathway, which led to upregulating NF-kB and MyD88. The MyD88 pathway plays an essential role in regulating the antiviral immunological response, as discussed earlier.

Our data showed that treatment with Rintatolimod increased the expression of MHC class I and II genes significantly. All class I genes were increased in a gradient fashion. The HLA-B gene was increased more than 20 times after treating with Rintatolimod. Cytokine and chemokine production was also significantly increased in pancreatic cancer cells treated with Rintatolimod. CXCL10 was increased more than 200 times after Rintatolimod treatment. All these cytokines are important in monocyte and macrophage activation. They are also known to act as chemoattractants for T cells, NK cells, and dendritic cells. Interestingly, treating with Rintatolimod did not induce the expression of CXCL12, which plays an important role in increasing the metastatic abilities of cancerous cells (Figure 2). Taken together, Rintatolimod induces an innate and an adaptive immune response in pancreatic cancer cells. The secreted cytokines attract immune cells including macrophages, NK cells, T cells, and B cells. The innate and adaptive immune systems play an important role in the fight against cancer and against SARS-CoV-2 or any other viral infections. Importantly, treating with Rintatolimod did not stimulate the metastasis-related genes, which indicates that Rintatolimod might be a safe drug to treat cancer patients who suffer from a viral infection, including COVID-19 infection, at an early stage to prevent further deterioration. This is in contrast to dexamethasone, which is only effective at an advanced stage of the infection in patients hospitalized with COVID-19; the use of dexamethasone decreases mortality only among those who are receiving either invasive mechanical ventilation or oxygen [11].

Rintatolimod could probably be co-administrated with remdesivir, which also inhibits multiplying the viral genome by inhibiting the viral RNA-dependent RNA polymerase. In cancer patients, PD-1 agonists are currently being investigated in combination with Rintatolimod because the combination could enhance the T cell function even further.

Importantly, treatment with Rintatolimod did not upregulate the expression of TGFβ1 (Figure 5). This is an important finding because TGFβ1 protein induces chronic fibrosis in the lung [42], which was reported in COVID-19 infected patients. Therefore, treating with Rintatolimod may help mitigate lung fibrosis. Moreover, treatment with Rintatolimod significantly upregulated the expression of various molecules that mediate the adhesion of immune cells to blood vessels like VCAM1 and TGFβ2 and many other integrin genes that are associated with increased blood vessel formation (Figure 5). The latest information from COVID-19 autopsies reported that patients suffer from congested, edematous, and widened blood vessels with infiltration of monocytes and lymphocytes [43]. Therefore, treating with Rintatolimod may increase angiogenesis and the infiltration of immune cells to the injured tissue.

As reported in our previous study, infecting BxPC-3 cells with Newcastle Disease virus (NDV) also activated the interferon signaling pathway [17]. However, the activation by Rintatolimod seemed to be much more effective. A report on an in vitro study using Rintatolimod against human coronavirus OC43 described that Rintatolimod was the only anti-viral drug that was highly effective at achievable human doses. In another study, Rintatolimod was shown to be an effective antiviral drug against SARS-CoV-1 in mice [26]. Rintatolimod has a well-developed safety profile based on more than 100,000 IV doses administrated to humans [24,26]. In addition, Rintatolimod is a well-tolerated drug that did not show severe adverse events when it was given to pancreatic cancer patients (data not shown).

Recently, the viral entry gene ACE2 was detected in nasal cells by using single-cell RNA-sequencing data from multiple tissues from healthy human donors [44]. Considering that the primary SARS-CoV-2 transmission is through infectious droplets and the release of the virus does not require cell lysis, it might exploit existing secretory pathways in nasal goblet cells sustained at a presymptomatic stage. Since the TLR3 receptor is highly expressed in the nasopharyngeal epithelial cells, Rintatolimod administered intranasally could be highly effective in limiting the spread and further deterioration of infected patients.

We did not investigate the effect of Rintatolimod on epithelial cells infected with SARS-CoV-2. In addition, the effect of Rintatolimod was limited to HPAC cells, without studying the synergetic effect on immune cells. Therefore, we think a clinical trial is essential to investigate the synchronized effect of the promising treatment.

## 5. Conclusions

In conclusion, by targeting TLR3, Rintatolimod stimulates HPAC cells to produce various cytokines and chemokines that reflect the activation of the adaptive immune system. Rintatolimod increases the expression of RNase L enzyme that degrades viral and cellular RNA. It activates the innate immune system by inducing the interferon signaling pathway. Taken together, Rintatolimod should be considered in the treatment regimens of cancer patients who suffer from SARS-CoV-2 infection.

## Figures and Tables

**Figure 1 cancers-13-02896-f001:**
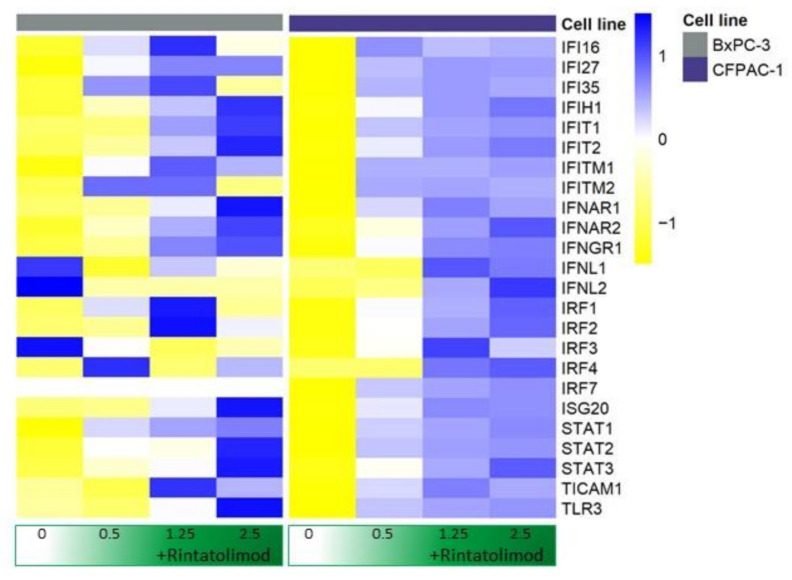
Rintatolimod stimulated interferon production and activated the interferon signaling pathway in HDAC cells. Heat maps of genes involved in the interferon signaling pathway. The linear values of the gene expression were used to generate the heat maps. The expressions were gene-wise scaled and are displayed as colors ranging from blue (high expression) to yellow (low expression) as shown in the key. Cells were treated with various concentrations of Rintatolimod displayed as the intensity of green color from dark green (high dose) to white (PBS control).

**Figure 2 cancers-13-02896-f002:**
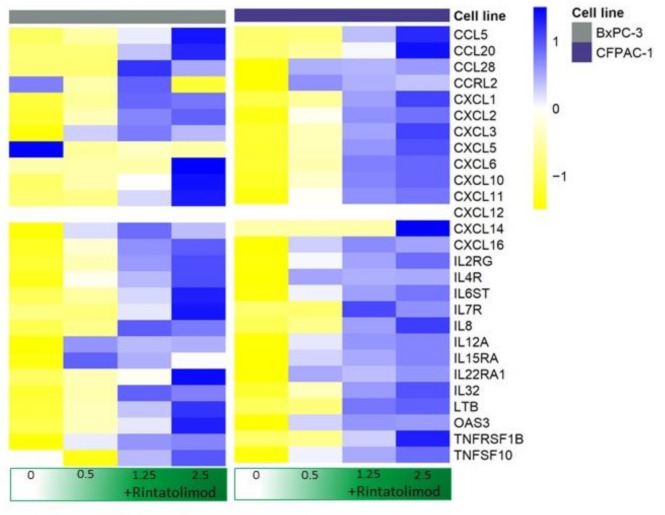
Rintatolimod produced immunomodulatory activity in HPAC cells. Heat maps of cytokines and interleukins that were differentially regulated as an effect of Rintatolimod. The expressions were gene-wise scaled and are displayed as colors ranging from blue (high expression) to yellow (low expression) as shown in the key. Cells were treated with various concentrations of Rintatolimod displayed as the intensity of green color from dark green (high dose) to white (PBS control).

**Figure 3 cancers-13-02896-f003:**
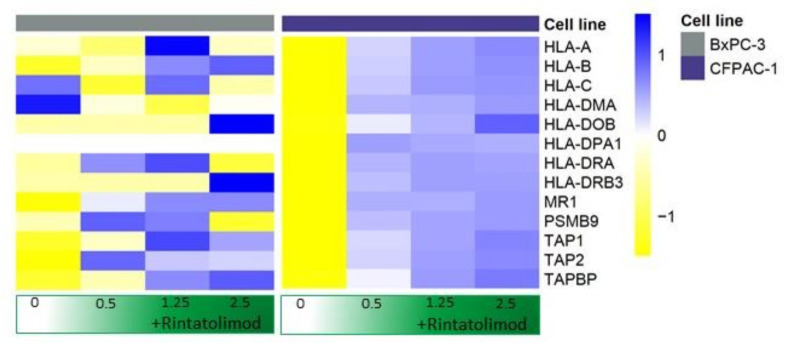
Rintatolimod induced the expression of MHC class I and II histocompatibility in HPDA cells. Heat maps of HLA-genes that were differentially regulated as an effect of Rintatolimod. The expressions were gene-wise scaled and are displayed as colors ranging from blue (high expression) to yellow (low expression) as shown in the key. Cells were treated with various concentrations of Rintatolimod displayed as the intensity of green color from dark green (high dose) to white (PBS control).

**Figure 4 cancers-13-02896-f004:**
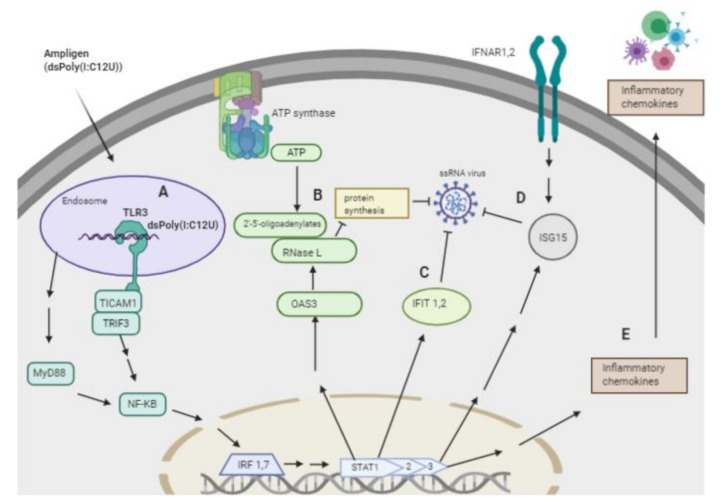
Schematic presentation of the overexpressed genes and the antiviral pathways activated by Rintatolimod. Rintatolimod (Ampligen; dsPoly (I: C12U)) enters the cells and targets the TLR3 in the endosome to start a cascade of actions. A—Stimulation of TLR3, which activates TICAM1 and TRIF3 that are involved in innate immunity against pathogens. Activation of TICAM1 and MyD88 mediate NF-kB expression that subsequently activates transcription factors such as IRF1 and IRF7. Interferon Regulatory Factors activate STAT1, 2, and 3 that stimulate the innate and acquired immune responses, mediate cellular responses to interleukin, and regulate inflammatory responses to infection. B—Activation of STAT1 and 2 upregulate the expression of OAS3, which induces the synthesis of 2′,5′-oligoadenylate (2–5 A) from ATP upon binding of dsRNA. 2–5 A activates RNase L, causing endonucleolytic cleavage of viral ssRNAs and blocking virus replication. C—TLR3 activation induces the expression of members of the IFIT family. The upregulation of IFIT1 and IFIT2 result in restriction of ssRNA virus replication in the cytosol. In addition, the IFIT family restricts virus replication through an alternation of protein synthesis and the ability of an antiviral protein to bind viral RNA directly. D—Activation of STAT 1, 2, and 3 activate the expression of interferon stimulatory genes (ISGs) like ISG15, which induces the antiviral state in the cytosol. ISG15 acts on ssRNA and directly degrades viral RNA. Treatment with Rintatolimod increased the expression of IFNAR1 and IFNAR2, which recognize interferon and activate JAK/STAT signaling pathways. E—Activation of STAT signaling pathways leads to the production of various cytokines, which results in the activation of the adaptive immune system. The pathway image was created with biorender.com.

**Figure 5 cancers-13-02896-f005:**
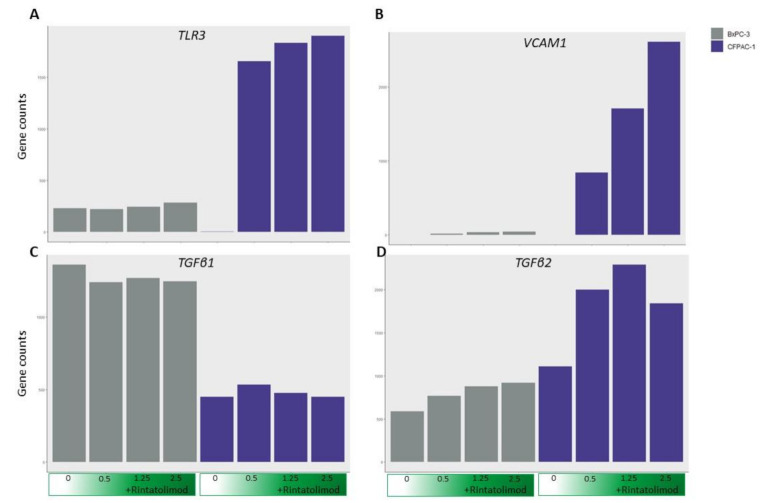
Treating cancer cells with Rintatolimod did not upregulate genes involved with fibrosis. The expression of TLR3 (**A**) was upregulated in CFPAC1 cells after treating with Rintatolimod. The expression of VCAM1 (**B**) and TGFβ2 (**D**) was upregulated by Rintatolimod. However, no upregulation of TGFβ1 (**C**) was observed after treating with Rintatolimod in both cell lines.

**Table 1 cancers-13-02896-t001:** Upregulated antiviral proteins in BxPC-3 cells that were treated with Rintatolimod.

Identified Proteins	Index Number	Associated Gene	Number of Peptides ^#^			
			CN *	2.5 **	1.25 **	0.5 **
Probable ATP-dependent RNA helicase DDX60	Q8IY21	*DDX60*	0	27	21	11
2′–5′-oligoadenylate synthase-like protein	Q15646	*OASL*	0	7	4	3
Single-stranded DNA-binding protein	P42566	*SSBP1*	0	4	5	1
Epidermal growth factor receptor substrate 15	P42566	*EPS15L1*	0	3	3	3
Probable ATP-dependent RNA helicase DDX58	O95786	*DDX58*	0	3	3	1
Transcription elongation factor SPT6	Q7KZ85	*SUPT6H*	0	3	3	1
HLA class I histocompatibility antigen, alpha chain E	P13747	*HLA-E*	0	3	2	2
Interferon regulatory factor 3	Q14653	*IRF3*	0	1	0	2
Interferon-induced protein with tetratricopeptide repeats 3	O14879	*IFIT3*	0	2	1	0

* Control Group (no treatment with Rintatolimod). ** Rintatolimod concentration (mg/mL) that was used in the experiment compared to the control group. ^#^ Numbers represent the number of unique peptides belonging to a specifically identified protein in each sample.

## Data Availability

The data presented in this study are available on request from the corresponding author. The genomic data are not publicly available due to privacy issues. The mass spectrometry proteomics data have been deposited to the ProteomeXchange Consortium via the PRIDE [42] partner repository with the data set identifiers PXD026116 and 10.6019/PXD026116.

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
