# Peer review of "Rintatolimod Induces Antiviral Activities in Human Pancreatic Cancer Cells: Opening for an Anti-COVID-19 Opportunity in Cancer Patients?"

_cancers, 2021, doi:10.3390/cancers13122896_

Round 1
Reviewer 1 Report
The manuscript was prepared very well. The introduction section justifies the purpose of the study. I congratulate the authors for the preparation of the manuscript
The subject of the review is certainly interesting and of relevance to the field, as there is a lack of potent measures to address COVID-19. However, there are some concerns, in part important, so the review articles need revision, see below.
Introduction
- The article would benefit from more information on anti-SARS-CoV-2 immune responses, i.e., the cell types and cytokines involved and the approximate time frame, in the introduction
- These clinical settings, the types of infection and the effects of Rintatolimod should be described in more detail, as they are central to the hypotheses put forward by the authors that Rintatolimod could be beneficial in COVID-19.
Materials and Methods
The methodology is perfectly described and carried out
Results
- The tables and the text describing them do not require any input, it is the strongest part of this study.
Discussion
- With what other drugs could Rintatolimod be co-administered?
- the most effective treatment in COVID-19 is PROBABLY Dexamethasone for COVID and cancer. What does Rintatolimod add that Dexamethasone does not? please clarify this question.
- Include a limitations section.
Author Response
The subject of the review is certainly interesting and of relevance to the field, as there is a lack of potent measures to address COVID-19. However, there are some concerns, in part important, so the review articles need revision, see below.
We thank the reviewer for the positive opinion about the way we prepared the manuscript and the interest in our work. We are very pleased with this opinion. Here are the answers we prepared for the comments/question.
Introduction
- The article would benefit from more information on anti-SARS-CoV-2 immune responses, i.e., the cell types and cytokines involved and the approximate time frame, in the introduction.
A paragraph about the host immune response after being infected with SARS-CoV-2 was added to the introduction, Page: 3, Line: 55-66.
- These clinical settings, the types of infection and the effects of Rintatolimod should be described in more detail, as they are central to the hypotheses put forward by the authors that Rintatolimod could be beneficial in COVID-19.
These points were addressed in the introduction, Page: 3-4, Lines: 94-102 &110-114 & 135-136.
Materials and Methods
The methodology is perfectly described and carried out
Results
- The tables and the text describing them do not require any input, it is the strongest part of this study.
We thank the reviewer for the encouraging opinion.
Discussion
With what other drugs could Rintatolimod be co-administered?
We thank the reviewer for this important point.
Rintatolimod could probably be co-administrated with Remdesivir which also inhibits multiplying of the viral genome by inhibiting the viral RNA-dependent, RNA polymerase. In cancer patients, PD-1 agonist is currently investigated in combination with this since this could enhance the T cell function even further. This part was added to the discussion, Page: 11, Lines: 318-321.
- The most effective treatment in COVID-19 is PROBABLY Dexamethasone for COVID and cancer. What does Rintatolimod add that Dexamethasone does not? please clarify this question.
We thank the reviewer for asking this question.
Rintatolimod might be a safe drug to treat cancer patients who suffer from a viral infection, including COVID-19 infection at an early stage to prevent further deterioration. This in contrast to dexamethasone, which is only effective at an advanced stage of the infection In patients hospitalized with Covid-19, the use of dexamethasone decrease mortality only among those who were receiving either invasive mechanical ventilation or oxygen [RECOVERY Collaborative Group, et al, Dexamethasone in Hospitalized Patients with Covid-19. N Engl J Med. 2021 Feb 25;384(8):693-704].
This part was added to the discussion, Page: 10, Lines: 313-317.
- Include a limitations section.
The limitation of the study is described on Page: 11, Lines: 351-354.
Reviewer 2 Report
In this submission, Mustafa et al. showed that Rintatolimod induces an antiviral effect in HPACs by inducing RNase L- dependent and independent pathways of the innate immune system. Treatment with Rintatolimod activated the interferon signaling pathway, leading to the overexpression of several cytokines and chemokines in epithelial cells. This is an interesting study, however, there are a few unclear mechanisms which I have listed below. I recommend this paper to be accepted with subject to major revisions.
- Line 48-49: it needs a references and also full names of various terms used here rather than abbreviations. I suggest adding a citation here, relevant to these sentences - https://doi.org/10.1016/j.nantod.2020.100962
- Line 54: 'Cancer and its treatments can lead to poor nutrition in patients which affect the immune system'. This sentence needs a bit more discussion on limited nutrition and immune system.
- Line 57: '....... studies that took most high-risk factors'. please specify these risk factors.
- Line 61: '..... an increased vulnerability in cancer patients for SARS-CoV-2 with a subsequent poor outcome'. This sentence seems incomplete. why is this poor outcome and what are key factors associated with resistance and poor outcomes? please elaborate.
- Line 65: when authors describe the efficacy of different drugs or reproposed drugs, they should classify them along with their original disease mechanism and why such drugs have been proposed for the treatment of COVID-19. Such insights will be helpful for readers to understand the merits/demerits of drugs and their corresponding therapeutic efficacies.
- Line 110: how did they choose three different concentrations of Rintatolimod: 0.5, 1.25, and 110 2.5 mg/ml?
- Why did they only perform cell culturing and treatment experiments twice? There should be a minimum three independent experiments. did they perform both experiments independently? if yes, they should specify this and these details should be added in figure captions as well.
- On which basis did they choose these cell lines? this should be explained and justified in the relevant section.
Author Response
In this submission, Mustafa et al. showed that Rintatolimod induces an antiviral effect in HPACs by inducing RNase L- dependent and independent pathways of the innate immune system. Treatment with Rintatolimod activated the interferon signaling pathway, leading to the overexpression of several cytokines and chemokines in epithelial cells. This is an interesting study, however, there are a few unclear mechanisms which I have listed below. I recommend this paper to be accepted with subject to major revisions.
We thank the reviewer for having an interest in our manuscript. We provide answers to the questions below.
- Line 48-49: it needs a references and also full names of various terms used here rather than abbreviations. I suggest adding a citation here, relevant to these sentences.
We thank the reviewer for this remark. We would like to highlight that full names were used prior to the abbreviations in lines 46-47.
- Line 54: 'Cancer and its treatments can lead to poor nutrition in patients which affect the immune system'. This sentence needs a bit more discussion on limited nutrition and immune system.
the sentence was adjusted.
“Cancer patients often experience weight loss, or even cachexia in severe cases. Cancer and its treatments can lead to poor nutrition in patients. Undernourishment weakens the quality of life, therapeutic response and affects the immune system.” Page: 3, Lines: 68-71 - Line 57: '....... studies that took most high-risk factors'. please specify these risk factors.
We apologies if the sentence is not clear. The high-risk factors were mentioned in the sentence before the one the reviewer is referring to. Namely: age, sex, and health condition, such as cardiovascular diseases and cancer are linked to the severity of COVID-19 infections. Page: 3, line: 71. - Line 61: '..... an increased vulnerability in cancer patients for SARS-CoV-2 with a subsequent poor outcome'. This sentence seems incomplete. why is this poor outcome and what are key factors associated with resistance and poor outcomes? please elaborate.
We thank the reviewer for this point. We clarified the text and elaborated to cover more details.
“An evidence collected from national and international cancer registries showed that patients with cancer infected with SARS-Cov-2 have a higher probability of death compared with patients without cancer [16]. The key factors associated with the poor outcome are common factors like: age, male sex, smoking history, number of comorbidities, and poor performance status.” Page: 3, Lines: 78- 82.
- Line 65: when authors describe the efficacy of different drugs or reproposed drugs, they should classify them along with their original disease mechanism and why such drugs have been proposed for the treatment of COVID-19. Such insights will be helpful for readers to understand the merits/demerits of drugs and their corresponding therapeutic efficacies.
We thank the reviewer for this point. We added the explanation to the text. Page: 3-4, Lines: 94-102 &110-114 & 135-136.
- Line 110: how did they choose three different concentrations of Rintatolimod: 0.5, 1.25, and 110 2.5 mg/ml?
The tested concentrations of Rintatolimod were chosen because they reflect the concentrations that were admitted to pancreatic cancer patients in our clinical work to treat patients with pancreatic cancer (in preparation). We added the reasons to the M&M, Page: 4, Line:146-148.
- Why did they only perform cell culturing and treatment experiments twice? There should be a minimum three independent experiments. did they perform both experiments independently? if yes, they should specify this and these details should be added in figure captions as well.
We apologies for the confusion. The cell cultures/ cell counts and observation studied were performed three times. However, culturing to obtain –omics data was done twice using 4 different concentrations. The methods we used are very sensitive, specially NanoString, but the cost is relatively high. We adjusted the text according to the reviewer's advice. Page: 4-5, Lines: 148-150.
- On which basis did they choose these cell lines? this should be explained and justified in the relevant section.
We thank the reviewer for this important question. The cell lines were chosen because they are one of the most studied cells in our laboratory. We currently are performing an extensive –omics analysis for the used cells and for other cell lines to be able to publish more comprehensive results. We try to stratify the cell lines based on their expression of TLR-3. The reasoning was added to the text, Page: 4, Line: 142-144.
Round 2
Reviewer 1 Report
The authors have improved the manuscript by incorporating all suggestions.
I have nothing more to add
Reviewer 2 Report
I am pleased to recommend the revised version of the manuscript for publication in Cancers.